# Insertional Inactivation of *Prevotella intermedia* OxyR Results in Reduced Survival with Oxidative Stress and in the Presence of Host Cells

**DOI:** 10.3390/microorganisms9030551

**Published:** 2021-03-07

**Authors:** Mariko Naito, B. Ross Belvin, Mikio Shoji, Qin Gui, Janina P. Lewis

**Affiliations:** 1Department of Microbiology and Oral Infection, Graduate School of Biomedical Sciences, Nagasaki University, Nagasaki 852-8588, Japan; mnaito@nagasaki-u.ac.jp (M.N.); m-shoji@nagasaki-u.ac.jp (M.S.); 2Philips Institute for Oral Health Research, Virginia Commonwealth University, Richmond, VA 23298, USA; belvinbr@vcu.edu (B.R.B.); qingui419@gmail.com (Q.G.); 3Department of Microbiology and Immunology, Virginia Commonwealth University, Richmond, VA 23298, USA; 4Department of Biochemistry, Virginia Commonwealth University, Richmond, VA 23298, USA

**Keywords:** *Prevotella intermedia*, mutagenesis, oxidative stress, host–pathogen interaction

## Abstract

One of the most abundant bacteria in the subgingival pockets of patients with bleeding following mechanical periodontal therapy is *Prevotella intermedia.* However, despite its abundance, the molecular mechanisms of its contribution to periodontal disease are not well known. This is mainly due to the lack of genetic tools that would allow examination of the role of predicted virulence factors in the pathogenesis of this bacterium. Here, we report on the first mutant in the *P. intermedia* OMA14 strain. The mutation is an allelic exchange replacement of the sequences coding for a putative OxyR regulator with *ermF* sequences coding for the macrolide–lincosamide resistance in anaerobic bacteria. The mutant is severely impaired in its ability to grow with eukaryotic cells, indicating that it is an important target for interventional strategies. Further analyses reveal that its ability to grow with oxidative stress species, in the form of hydrogen peroxide and oxygen, is severely affected. Transcriptome analysis reveals that the major deregulated genes code for the alkylhydroperoxide reductase system, AhpCF, mediating protection from peroxide stress. Moreover, genes coding for Dps, CydA and Ftn are downregulated in the mutant strain, as further verified using qRT-PCR analysis. In conclusion, we succeeded in generating the first *P. intermedia* mutant and show that the OxyR-deficient strain is unable to survive with a variety of host cells as well as with oxidative stress.

## 1. Introduction

*Prevotella intermedia* is a highly abundant bacterium in the oral cavity. As shown in our recent publication, the genus *Prevotella* was one of the most abundant bacteria in biofilm samples collected from the subgingival pockets of patients with periodontal disease [1], as well as being prevalent in the salivary microbiome [2]. It is a black pigmented, anaerobic, Gram-negative bacterium, associated with periodontal disease, a destructive disease of the supporting structures of the teeth [3]. Lopez et al. found a high prevalence of *Porphyromonas gingivalis* and *P. intermedia* in adult periodontitis lesions. Although *P. intermedia* is also found at healthy sites [4,5], the profile of degradative enzymes produced by the organism varies depending on the site at which it is present [6]. This suggests that this organism alters the enzyme profile under certain conditions to promote the progression of periodontitis. In addition to periodontitis, *P. intermedia* has also been associated with endodonthic infections [7,8], as well as being found in extraoral sites, e.g., NOMA (cancrum oris, an infection that destroys the oro-facial tissues) lesions [8,9]. Lastly, studies suggest that chronic infections including those associated with periodontitis increase the risk of systemic diseases such as coronary heart disease and preterm delivery of low birthweight infants [10]. Genco et al. found periodontopathogens, including *P. intermedia,* in atherosclerotic plaques [11], and the Offenbacher group [12] demonstrated the significantly higher prevalence of positive fetal IgM to *P. intermedia* for preterm as compared full-term infants. Importantly, evidence of its contribution to systemic conditions, such as the induction of severe bacteremic pneumococcal pneumonia, has been reported [13]. Finally, one of the most clinically relevant findings is the presence of *P. intermedia/nigrescens* in nearly all human subjects exhibiting bleeding upon probing following mechanical periodontal therapy and in a high proportion of sites [14]. This is in contrast with the presence of the periodontopathogen *Porphyromonas gingivalis* that was found in a high proportion of subjects but in a low proportion of bleeding sites. Such findings warrant more detailed investigation of the molecular mechanisms of *P. intermedia* interaction with the host in the oral cavity.

Adaptation to the host’s environment is one of the most important factors allowing for persistence for all bacteria, including *P. intermedia.* Two major environmental factors play a role in the oral cavity: the concentrations of iron and oxygen (both atmospherically converted to endogenous through metabolic processes and oxidative stress species released by the host’s innate immune response) [15]. We have previously shown that iron levels regulate the expression of several proteins, including the *hmu* operon-encoded HmuY-like and HmuR-like proteins and a thioredoxin-like protein [16]. The *hmu* operon was also shown to be regulated in response to iron in *P. gingivalis* [17,18]. Further information on the HmuY homologs expressed by *P. intermedia* (termed PinO and PinA), produced at higher levels under iron starvation, was reported recently [19]. The molecular mechanisms of this regulation, however, remain unclear. The main obstacle in studying *P. intermedia* is the inability to manipulate it genetically. Here, we succeeded in generating a deletion mutant in the putative OxyR regulator in *P. intermedia.* The OxyR regulator, a member of the LysR family of regulators, was first characterized as a positive regulator of a number of peroxide stress-response-encoding genes in *E. coli* and demonstrated to be required for the survival of the bacterium with oxidative stress [20,21]. It later was found in a number of Gram-negative as well as Gram-positive bacteria and also shown to be required for protection of the bacteria from various forms of oxidative stress [20,22,23,24,25]. It is noteworthy that it also was reported in bacteria belonging to the Bacteroidetes family, such as *Bacteroidetes thetaiotamicron*, *Bacteroides fragilis, P. gingivalis* and *Tannerella forsythia* [22,26,27,28,29]. The latter two are oral bacteria and major periodontal pathogens. *P. gingivalis* also forms black pigmented colonies, similar to those observed for *P. intermedia* [30,31]. OxyR was shown to play a role in the adaptation of *P. gingivalis* to peroxide stress [29,32,33] and we hypothesized that it may also be required for a similar purpose in *P. intermedia.* This is the first mutant generated in this species and our method is expected to pave the way towards determining the *P. intermedia* molecular mechanisms of adaptation to the environment as well as virulence.

## 2. Materials and Methods

### 2.1. Bacterial Strains and Growth Conditions

For our study, we used *Prevotella intermedia* OMA14 strain. This is a clinical isolate that has recently been sequenced [34]. It was isolated and maintained on TS agar plates (40 g/L tryptic soy agar base supplemented with 10 g/L of yeast extract, 1 g/L cysteine, 5 µg/mL hemin and 0.5 µg/mL menadione) and broth cultures were grown in enriched BHI medium (37 g/L BHI, 2.5 g/L yeast extract, 5 µg/mL hemin and 0.5 µg/mL menadione). For phenotypic analyses, *P. intermedia* parental and mutant strains (parental OMA14 (V3203) as well and OxyR-deficient strain generated in this study (V3147)) were maintained anaerobically on sheep blood agar (TSA II, 5% sheep blood). *E. coli* S17-1 was maintained on LB agar plates supplemented with 100 µg/mL trimethoprim and 50 µg/mL streptomycin. Antibiotics were used at the following concentrations: ampicillin (Ap; 100 μg/mL for *E. coli*), erythromycin (Em; 10 μg/mL for *P. intermedia*) and gentamycin (Gm; 100 μg/mL for *P. intermedia*). For iron-replete conditions, bacteria were grown in BHI media supplemented with 5 μg/mL hemin. For iron-deplete conditions, the BHI broth was supplemented with 150 µM of dipyridyl (DP) and 5 μg/mL PPIX (to compensate for the non-iron component of hemin) [16]. Cells were harvested in mid-log phases (at an OD_660_ of approx. 0.5).

### 2.2. Generation of P. intermedia Mutant Strain

*Plasmid construction.* The plasmids and oligonucleotides used in mutant generation are listed in Appendix A, respectively. A 1.4-kb PCR-amplified *SacB* gene from pDNR-Dual (primers: sacB-N-*Bgl*II + sacB-C-*BamH*I) was cloned into pUC118 at the *Hinc*II site, resulting in pUC118-sacB. The PCR-amplified fragment of *Porphyromonas gulae* catalase promoter from pKD954 (primers: p6-34-F-BglII + p6-34-R-*BamH*I) was cloned into pUC118 at the *Hinc*II site, resulting in pUC118-Pcat. The BglII-BamHI fragment obtained from pUC118-Pcat was inserted into the BglII site of pUC118-sacB, resulting in pUC118-Pcat-sacB. The *ermF* 1.1 kb BamHI fragment was obtained from pKD718 [35].

DNA regions up- and downstream of the *oxyR* gene were amplified by PCR from *P. intermedia* OMA14 chromosomal DNA with two pairs of primers (oxyR-up-F-EcoRI + oxyR-up-R-BglII; oxyR-dw-F-BglII + oxyR-dw-R-BamHI, where “up”, “dw”, “F” and “R” indicate upstream, downstream, forward and reverse, respectively). The up- and downstream fragments of *oxyR* gene were mixed and amplified (primers: oxyR-up-F-EcoRI + oxyR-dw-R-BamHI). The subsequent target fragments were ligated into pBSSK (pND0001). In the resulting plasmid, the *ermF* 1.1 kb *Bam*HI fragment was inserted into the *Bgl*II site, after which the 1.8-kb Pcat-sacB *Bgl*II-*Bam*HI fragment was inserted into the BamHI site to yield the targeting cassettes (pND0002). The *Not*I-*Xho*I fragment of the targeting cassette was then ligated into the *Not*I-*Xho*I site of pTCB to yield conjugation plasmids for mutagenesis (pND0003) (Figure 2A).

*Mutagenesis of P. intermedia.* Conjugal transfer of the targeting plasmid between *E. coli* and *P. intermedia* was performed. Targeting vectors were transformed into *E. coli* S17-1, as previously described [36]. The transformed *E. coli* was pre-cultured in 5 mL LB medium supplemented with Ap (LB-Ap) at 37 °C for 16 h with gentle shaking. The full-grown culture (1 mL) was added to 25 mL LB-Ap and was further cultured for 3 h at 37 °C. The recipient, *P. intermedia* OMA14*,* was pre-cultured in 5 mL enriched BHI medium for 16 h at 39 °C under anaerobic conditions. All of the grown recipient culture was added to 35 mL enriched BHI medium and was further cultured at 39 °C for 6 h under anaerobic conditions. Prior to conjugation, donor *E. coli* cells were harvested by centrifugation at 800× *g* for 20 min at 4 °C. The supernatant was decanted, and the recipient culture was added to the donor cells. Cells were then centrifuged at 3300× *g* for 10 min. The supernatant was decanted, and the cells were suspended in the remaining supernatant and plated onto TS agar plates. After the plates were dried, conjugation culture was carried out aerobically at 37 °C for 2 h, and then anaerobically at 39 °C for another 15 h in an anaerobic chamber. All cells on the conjugation plate were scraped using a bacterial spreader and were then plated on TS agar plates supplemented with Gm and Em (TS-GE plate); cells were cultured in an anaerobic jar at 37 °C for 10 days. To eliminate the *sacB* gene and the plasmid backbone, the transformants were cultured on TS-GE plates supplemented with 7% sucrose for 5 days. The final mutant strains were termed V3147 (*oxyR::ermF*). Correct deletion of *oxyR* gene was verified by PCR (Figure 2C).

### 2.3. RNAseq Analysis of P. intermedia

Pellets of bacterial strains grown in iron-replete and iron-deplete conditions were used for the analysis. RNA was isolated using the RNeasy mini-kit (Qiagen, Hilden, Germany), following the manufacturer’s protocol. Any remaining DNA was removed using the DNA-free DNase kit (Ambion, Austin, TX, USA), following the manufacturer’s protocol. RNA quality was verified using agarose gel electrophoresis. RNA-seq library was then generated using the Ovation Complete Prokaryotic RNA-Seq DR multiplex kit (Nugen, Redwood City, CA, USA), following the manufacturer’s instructions. Quality of the libraries was verified using the bioanalyzer (Agilent Technologies, Santa Clara, CA, USA). The transcriptome-derived libraries were sequenced by the VCU’s Nucleic Acid Sequencing core using MiSeq Illumina Genome Analyzer. The resulting sequences were aligned to the reference genome of *P. intermedia* OMA14 using the CLC genomic workbench (CLC Bio, Qiagen, Hilden, Germany). To calculate the enrichment for the OxyR-dependent regulation, we divided the number of reads derived from the mutant to that derived from the parental strain for each position of the genome following normalization by aligned fragment per kilobase of transcript per million (FPKM). Similar bioinformatics work has been previously done in our laboratory [37,38].

### 2.4. Quantitative RT-PCR Analysis

Overnight cultures were inoculated in fresh BHI broth to an OD_660_ of 0.15. Cells were grown under anaerobic conditions until mid-log phase (~0.5–0.6 OD_660_ approximately 5–6 h) and then split. To expose them to oxygen, cells were placed in a shaking incubator under aerobic conditions for 30 min at 37 °C. For hydrogen peroxide exposure, two sequential doses were used. Hydrogen peroxide was added to cultures at a final concentration of 50 µM for 15 min at 37 °C under anaerobic conditions, after which hydrogen peroxide was added again (to a final concentration of 100 µM) and incubated for 15 min. A culture of bacteria was left in an anaerobic environment at 37 °C for 30 min to serve as an untreated control. After exposure, cells were spun down and washed with ice-cold PBS and stored at −20 °C. The RNeasy minikit (Qiagen) was used to purify RNA from cells and residual DNA was removed using the DNA-free DNase kit (ThermoFisher, Cambridge, MA, USA) via the manufacturer’s protocol. The cDNA was generated using the PhotoScript II cDNA synthesis kit (New England Biolabs, Beverly, MA, USA). Real-time qRT-PCR was performed using a SYBR green-based detection system on a Quant Studio 3 real-time PCR system using PowerUP Syber green master mix (ThermoFisher). The 16S ribosomal subunit was utilized as an endogenous control. All primers used for qRT-PCR analysis are listed in Appendix A.

### 2.5. Disc Diffusion Assay

Overnight cultures prepared in BHI broth were diluted to 0.2 OD_660_ and grown to mid-log phase (0.5–0.6 OD_660_) at 37 °C under anaerobic conditions. Cell cultures were normalized to an OD_660_ of 0.5 and 200 µL was spread onto TSA plates. Once plates were dry, a 10-mm filter disc was aseptically added to the plate. Directly to the disc, 10 µL of 5% or 10% H_2_O_2_ solution was added. Plates were incubated for 2–3 days and the size of the zone of exclusion was recorded.

### 2.6. Growth under Micro-Aerobic Conditions

Overnight cultures were diluted to an OD_660_ of approximately 0.15 in BHI broth pre-equilibrated in an atmosphere of 6% O_2_, 5% CO_2_, 5% H_2_ and 84% N_2_. Cultures were placed in airtight canisters and the atmosphere replaced using an Anoxomat anaerobic culture system to a final oxygen concentration of 6%. Cells were incubated to timepoints and growth assessed via measurement of OD_660_.

### 2.7. Outgrowth after Aerobic Exposure

Overnight cultures in BHI media were diluted to 0.15 OD_660_ and grown to 0.5 OD_660_ under anaerobic conditions. Cultures were then centrifuged and suspended in pre-warmed, anoxic BHI media. The cultures were incubated aerobically with vigorous shaking at 37 °C for 1 and 2 h. After O_2_ exposure, the OD_660_ was recorded and cell cultures were centrifuged. Cell pellets were suspended in pre-warmed anaerobic media to an OD_660_ of 0.15 and grown in the anaerobic chamber at 37 °C. After 24 h, growth was assessed via OD_660_.

### 2.8. Interaction of P. intermedia with Host Cells

Commercially available primary human umbilical vein endothelial cells (HUVECs) (Lifeline Technologies, Walkersville, MD, USA) were used in our studies. Cells were grown according to manufacturer’s instructions using cell-specific media in a 5% CO_2_ incubator. For our studies, cells were first grown to confluency in flasks and sub-cultured into 12-well tissue culture plates at approximately 50,000 cells per well and grown overnight. The next day, cells were placed in the anaerobic chamber and medium was replaced with pre-warmed anaerobic media. The cells were incubated for 4 h in anaerobic chamber prior to infection. Overnight cultures of *P. intermedia* were diluted to 0.15 OD_660_ and grown to log phase (approximately 0.5–0.6 OD_660_) and spun down and resuspended in warm host cell media. Wild-type *P. intermedia* OMA14 (V3203) or the *oxyR-*deficient mutant strain V3147 were added to host cells at a multiplicity of infection (MOI) of 100:1. The bacteria–cell mixture was incubated for 1 h in anaerobic chamber at 37 °C. For total interaction analysis, infected cells were washed in an anaerobic chamber 3 times using 1 mL/well of PBS. Washed cells were lysed with 1% saponin in BHI media to release bacteria and the mixture was serially diluted using anaerobic BHI medium. The diluted mixture was then plated on TSA blood agar plates and incubated for 5–8 days to allow for colony formation. The number of surviving bacteria was determined by counting the CFU/mL of each infection. To determine survival of internalized bacteria, following 1 h infection, cells were treated with 400 µg/mL metronidazole and 300 µg/mL gentamicin to kill extracellular bacteria. Following 1 h of antibiotic treatment, infected cells were washed, lysed, and CFU determination was carried out as described above.

### 2.9. Statistical Analysis.

All experiments were performed at least three times. Data in graphs are presented as means and error bars are presented as the standard error (SE) between replicates. The statistical analysis was performed via paired T-test for sample means of wild-type and mutant. A *p* value of <0.05 is considered statistically significant.

## 3. Results

### 3.1. Bioinformatics Analysis of P. intermedia’s OxyR Ortholog

We have selected a gene, PIOMA14_RS00365 (old locus tag: OMA14_I_0073), present on the large genome (GenBank: NZ_AP014597) coding for a predicted peroxide-inducible activator and possible OxyR homolog. The gene is 924 bp and codes for a 308-aa protein. This protein is predicted to play a role in the adaptation of *P. intermedia* to environmental stress as it is annotated as a peroxide activated regulator. Oxidative stress is the major variable in the oral cavity where the bacterium has to adapt to varying levels of atmospheric oxygen as well as to the oxidative stress released by the host immune defense mechanisms. Under these conditions, an effective regulator of the oxidative stress defense mechanisms is expected to play a major role in the persistence and survival of the bacteria. Thus, we selected the homolog of the OxyR regulator as it was consistently shown to play a role in adaptation to oxidative stress in variety of bacteria. Sequence comparison revealed that the most analogous OxyR proteins compared to *P. intermedia* were *P. dentalis* (72%), *P. melaninogenica* (73%), *P. oris* (69%), *B. thetaiotaomicron* (61.0%) and *B. fragilis* (61%) (Figure 1A). As there are many OxyR-like genes sequenced from various bacteria, we next selected only proteins that were thoroughly analyzed (including crystallographic studies) for our comparison studies. As shown in Figure 1B, the OxyR from bacteria belonging to the *Bacteroidetes* phylum, *P. gingivalis* (3HO7_A, 3UKI_A), shared 49% identity with the *P. intermedia* OxyR. There was 30% (4X6G_A) identity to the *P. aeruginosa* OxyR, 30% (1I6A_A) identity to the *E. coli* OxyR and 26% (6G1B_B) identity to the *Corynebacterium glutamicum* OxyR (Figure 1C). It is noteworthy that the conserved catalytic cysteines (designated as red asterisks in Figure 1B) were conserved in all the proteins that we analyzed in this study, implying similar mechanisms of action.

The genomic locus containing the gene encoding the *P. intermedia* OxyR protein is present on the larger chromosome of *P. intermedia* OMA14, NZ_AP014597 [34]. It is upstream of a gene coding for enolase and then two genes, *ahpCF,* coding for two subunits of the alkylhydroxyperoxide reductase, AhpC and AhpF (Figure 2B). The organization of the genomic loci is highly conserved in all *P. intermedia* strains sequenced so far (Appendix A). In *P. dentalis, P. melaninogenica* and *P. scopos*, the *oxyR* gene is transcribed divergently to *ahpCF*, and the *eno* gene is absent in the locus. Of note, in *P. dentalis*, the *ahpCF* locus is preceded by the *sod-*coding gene for superoxide dismutase (Appendix A).

### 3.2. Generation of P. intermedia Insertional Deletion Mutant Strain

For the preliminary experiment, ATCC strain *P. intermedia* 17 and 30 clinical isolates of *P. intermedia* were screened for the ability to receive plasmid DNA through conjugal transfer via the *Escherichia coli*-*Bacteroides* shuttle plasmid, pTCB. Transformants were obtained only from the clinical strain, OMA14. The complete genome sequence of this strain has previously been determined, and all genes have been identified and annotated [34]. To generate an insertional mutant for the *oxyR* gene, we constructed a targeting vector based on the *E. coli-Bacteroides* shuttle plasmid (Figure 2A). Briefly, a selection marker gene, *ermF*, was inserted between DNA fragments up- and downstream of the target gene in the plasmid containing the targeting construct. A counter-selection marker was also included to isolate clones that lost the plasmid DNA; the *Bacillus subtilis sacB* gene, which codes for the enzyme levansucrase, and is toxic in the presence of sucrose, was inserted into the targeting vector.

We first attempted mutagenesis in *P. intermedia* under anaerobic conditions, as was previously performed in *P. gingivalis*; only a few transformants were obtained. We then changed the experimental condition to an aerobic condition, which was adopted from conjugation in *Bacteroides uniformis* [39]. Under aerobic conditions, the viability of *P. intermedia* OMA 14 was maintained for 2 h, but quickly dropped after 3 h. The transfer efficiency was slightly improved when the mating culture was exposed to aerobic conditions for 2 h, followed by incubation in an anaerobic jar for 15 h. Furthermore, we found that a slight increase in the temperature of the recipient cell and mating cultures also improved transfer efficiency. Finally, we were able to consistently generate transformants at a frequency of 1×10^−10^ to 1×10^−11^. The erythromycin (Em)-resistant transformants were then cultured in the presence of sucrose. The resulting mutant clones were examined for genome organization by PCR analysis (Figure 2C). The loss of the transcript coding for *oxyR* was also verified through RNAseq (Figure 2D). Since, downstream of the *oxyR* gene, there is a gene coding for enolase, we also verified the transcriptional organization of the locus. As we had anticipated, the *oxyR-*specific transcripts were eliminated. No effect on the transcription of the downstream gene, *eno,* was observed, as evidenced by the RPKM for *eno* of 881 in the V3203 (wild-type strain) and RPKM for *eno* of 1309 in the V3147 (mutant strain), thus verifying the successful generation of the mutant strain (Figure 2D). Interestingly, we also observed that the *ahpCF* transcripts located downstream of the *eno*- specific transcripts were missing (Figure 2D).

### 3.3. OxyR Plays a Role in Oxidative Stress Homeostasis in P. intermedia

To gain insight into the spectrum of genes relying on the activity of OxyR, we performed preliminary assessment of the transcriptional profiles of the parental and mutant strains grown in iron-deplete and iron-replete conditions (Appendix A). We noted multiple genes with affected expression. Appendix A, listing the downregulated genes in the OxyR mutant (2-fold, *p* ≤ 0.1), shows that the *ahpCF* genes are drastically downregulated in the mutant strain compared to the wild type (378- and 153-fold, respectively). A significant but lesser regulation (18.3-fold) was observed for PIOMA14_RS09985 coding for the DNA starvation/stationary phase protection protein. Overall, 148 genes were downregulated more than twofold with *p* ≤ 0.1, indicating the extensive effect of OxyR deletion on adaptation mechanisms in *P. intermedia.* Similarly, there were also 88 genes upregulated in the OxyR-deficient mutant strain (Appendix A). The most highly upregulated genes were ones with hypothetical annotation.

We also examined the role of OxyR under iron-deplete conditions. As shown in Appendix A, a similar pattern of gene regulation was observed. The drastically downregulated expression of the *ahpCF* locus indicates that the *P. intermedia* OxyR primarily activates the expression of this locus under both iron-replete and iron-deplete conditions. Based on the above data, we have focused our attention on verification of the regulation of the *ahpCF* locus as well as the regulation of other known and putative ROS protection genes. In the *oxyR-*deficient strain, upregulation of *ahpC, ahpF* and *dps* was completely abolished under aerobic conditions as well as in response to H_2_O_2_ exposure (Figure 3A,B). However, the *cydA* and *ftnA* genes still displayed modest upregulation, indicating that these genes may be upregulated in a non-OxyR-dependent manner (Figure 3A,B). Exposure of wild-type *P. intermedia* OMA to aerobic conditions or H_2_O_2_ was sufficient to upregulate both the *ahpC* and *ahpF* alkyl-peroxidase genes, as well as genes coding for the putative iron-sequestering and protection proteins *dps* and *ftnA* (ferritin) (Figure 3B). The *cydA* gene, coding for a cytochrome d ubiquinol oxidase, was also upregulated in response to oxygen or ROS (Figure 3A,B).

### 3.4. P. intermedia Requires OxyR to Grow and Survive with Oxidative Stress and Oxygenated Conditions

Based on the gene regulation, we suspected that the *P. intermedia oxyR* mutant strain will have reduced growth and survival in response to oxygenation and ROS. The *oxyR* mutant strain exhibits sensitivity to H_2_O_2_ on disc diffusion plates (Figure 4A,B). There is no significant difference in the growth of the WT strain and mutant strain under anaerobic conditions (Figure 4C). However, the *oxyR* mutant exhibits a significant decrease in growth when exposed to oxygen. An atmosphere of 6% O_2_ can repress the growth of the *oxyR* mutant strain (Figure 4D). Furthermore, outgrowth after exposure to aerobic conditions with vigorous shaking severely reduces the capacity of the mutant strain to rebound and grow when moved back to anaerobic conditions (Figure 4E).

### 3.5. OxyR Is Required for P. intermedia Survival with Host Cells

To gain an insight into the ability of the mutant to survive in the presence of host cells, we used a cellular model by utilizing the primary human umbilical vein endothelial cells (HUVECs) [32,40]. The studies were done under anaerobic conditions, thus not interfering with the oxygen sensitivity of *P. intermedia.* As shown in Figure 5, we were able to recover a significant number of bacteria following 1 h exposure to host cells. Similarly, internalized *P. intermedia* was recovered in large numbers (Figure 5). However, a significant reduction was observed for the OxyR-deficient V3203 strain exposed for 1 h to HUVECs. Similarly, internalized V3203 bacteria were recovered from HUVECs in drastically reduced numbers when compared to the wild type. Thus, the above data demonstrate that the *P. intermedia* OxyR is indispensable for the survival of the bacterium with host cells.

## 4. Discussion

We successfully generated the first *P. intermedia* isogenic mutant through allelic replacement of the OxyR-coding sequences for that encoding the ErmF, providing resistance for macrolide–lincosamide in anaerobic bacteria [41]. The mutant strain exhibited reduced ability to survive with oxidative stress. It is noteworthy that the mutant has significantly reduced ability to survive with host cells, thus indicating that the *P. intermedia* OxyR could be a good target for the development of strategies aimed at reducing the fitness of the bacterium. We have also observed a reduced number of internalized bacteria, thus indicating that the bacterium deficient in the OxyR regulator has reduced ability to survive when internalized by host cells. Further studies examining the rates of microbial invasion as well as inhibitors of the host’s NADPH oxidase are needed to ensure that the oxidative stress tolerance is the main reason for the reduced recovery of the mutant strain from the host cells. However, our finding is highly relevant for clinical purposes as recurrent periodontitis is a major complication in the treatment of periodontal diseases and internalized bacteria are a reservoir for recurrent infection [42]. Thus, any strategy that would lead to a reduction in intracellular bacteria is also expected to be relevant in reducing the rates of recurrent periodontitis [43].

The most drastically downregulated genes in the OxyR mutant V3147 belonged to the *ahpCF* operon coding for the peroxide-reducing system. AhpC is the peroxidase catalytic component and AhpF is the AhpC-reactivating enzyme. The AhpCF system has been shown to play a role in scavenging microscale peroxide levels [44]. AhpC employs two conserved cysteine residues to reduce peroxide via formation of a disulfide bond. AhpF restores the catalytic cystines, utilizing NADH as an electron donor [45,46]. These enzymes also play a role in the early exponential growth phase and play a lesser role during the late exponential or stationary growth phases. It is of note that *P. intermedia* and its closely related *Prevotella* species do not code for superoxide dismutase, which is highly relevant for bacteria exposed to oxygen (Appendix A). We observed that the *ahpCF* operon was also significantly upregulated by exposure to oxygen and was required for the survival of *P. intermedia* exposed to oxygen. It is noteworthy that *P. intermedia* codes for glutathione peroxidase (GPX), which possibly plays a role in oxidative stress protection (Appendix A). However, *gpx* was not significantly regulated by exposure to oxygen. This possibly is due to the fact that the gene is already highly expressed in *P. intermedia.* RPKMs for *gpx1*and *gpx2* are 168.6 and 1953.2 in wild-type anaerobic conditions. These expression levels are 1.2- and 14.9-times that of *oxyR*. Expression of the *dps* gene, which encodes a DNA-binding protein expressed in starved cells, was also significantly downregulated in the OxyR mutant V3147 strain. Expression of dps has been shown to be affected by exposure to a variety of environmental stressors and suppression of *dps* expression in *P. gingivalis* and *B. fragilis* lacking OxyR [29,32,47]. Finally, reduced expression was observed for *cydA* and *ftnA* coding for cytochrome d ubiquinol oxidase and ferritin, respectively. However, some induction of these two genes was still observed in the presence of both oxygen and peroxide, thus indicating that other regulatory mechanisms mediate the expression of these genes.

The OxyR regulator is a LysR-like regulator that directly senses peroxide and forms intramolecular disulfide bonds that in turn lead to allosteric activation of the protein to enhance the transcription of antioxidant genes such as *ahpCF, katG, ccpA, dps, oxyS* [48,49,50,51]. In addition to its activator role, as listed for *E. coli,* it may also function as a repressor in other bacteria, such as *Neisseria gonorrhoeae* and *Burkholderia thalindensis* [23,52]. In *Neisseria,* OxyR acts in repression of catalase, the major anti-oxidative stress response mechanism [23]. However, as our results show, *P. intermedia* OxyR mainly functions in the activation of oxidative stress response mechanisms, resembling the function of OxyR in other members of the *Bacteroidetes* group.

The *P. intermedia* OxyR sequence has significant conformity with other members of the OxyR-like regulators. Besides being nearly identical to *oxyR* genes in other *Prevotella* species, it also is highly analogous to the OxyR in *B. fragilis* 638R and *B. thetaiotaomicron* VPI5482 [26]. The function of these proteins has been characterized, showing that they are indispensable for the upregulation of oxidative stress defenses [27,47]. The most analogous protein whose function and structure has been reported is that of the *P. gingivalis* OxyR [26]. However, despite the similarity between the proteins, there is a significant difference in the genomic organization of the loci encoding OxyR. *P. intermedia oxyR* is located adjacent to the *aphCF* locus; however, *P. gingivalis oxyR* are located on different genomic loci that are distant from each other.

For our work, we have used the *P. intermedia* strain OMA14 as the genome of this bacterium as determined and characterized in our earlier studies [34]. The core genes, including most of the virulence genes, were highly conserved. As regards the non-core genes, there was over 70% sequence conservation and the variations observed were due to the large number of different types of mobile elements. The annotated sequences facilitate the use of comprehensive genome-wide approaches such as RNAseq, ChIP-seq and proteomics. Mutagenesis of other strains of *P. intermedia* so far has not been successful. The efficiency of gene manipulation varying greatly from strain to strain is a common phenomenon. In fact, the efficiency of gene manipulation in *P. gingivalis* varies greatly from strain to strain [53]. In each strain (strain 17 and OMA14), a large number of genes related to DNA restriction modification and glycosyltransferase gene clusters are present that are strain-specific [34]. Such variations might cause differences in gene manipulation efficiency. We successfully deleted the gene coding for the putative OxyR. However, thus far, we have not been able to complement the mutation due to the lack of suitable selection markers. Moreover, since the efficiency of gene manipulation is still low, the creation of a marker-free type mutant strain has not been successful.

### RNAseq Accession Number

RNAseq data were deposited in the Gene Expression Omnibus (GEO) with the reference number GSE168003.

## Figures and Tables

**Figure 1 microorganisms-09-00551-f001:**
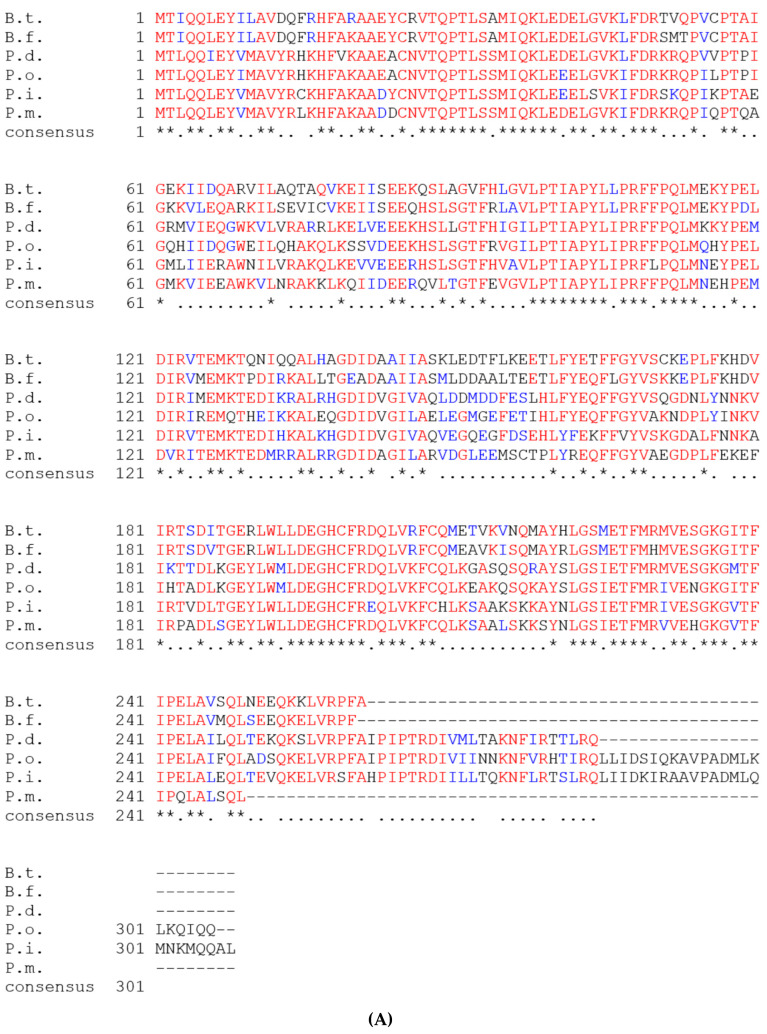
Bioinformatics analysis of *P. intermedia* OxyR. (**A**). Alignment of OxyR sequences from *P. intermedia* OMA14 (P.i.) to other members of the Bacteroidetes. Identity relative to *P. intermedia* OxyR is as follows: *P. dentalis*, 72%, *P. melaninogenica*, 73%, *P. oris*, 69%, *B. thetaiotaomicron,* 61% and *B. fragilis*, 61%. Amino acids are colored based on consensus to *P. intermedia* OMA14 OxyR (Red: identical AA conserved; Blue: similar AA conserved). (**B**) Alignment of OxyR sequences from *P. intermedia* OMA14 (P.i.) (this work), *P. gingivalis* (P.g.) (3HO7_A), *Vibrio vulnificus* (V.v.) (5B70_A), *Corynebacterium glutamicum* (C.g.) (6G1B_B), *Escherichia coli* (E.c.) (1I69_A), *Pseudomonas aeruginosa* (P.a.) (4XWS_A), *Neisseria meningitides* (N.m.) (3JV9_A). Legend of a consensus: asterisks denote identical amino acids and periods denote conserved amino acid substitutions. Red asterisks denote the active site cysteines. (**C**) Percent identity matrix for OxyR amino acid sequence for microbial species as in Panel (B). Protein alignments generated using ClustalW 2.1.

**Figure 2 microorganisms-09-00551-f002:**
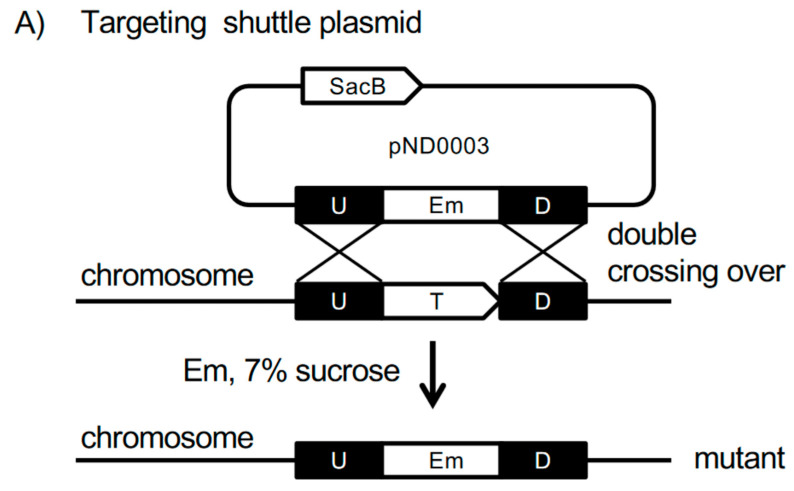
Generation of *P. intermedia* isogenic mutant. (**A**) Schematic diagram for insertional deletion mutagenesis in *Prevotella intermedia* OMA14 using *sacB* counterselection. The targeting shuttle plasmid harbored 0.6-kb fragments (U and D) homologous to the areas adjacent to the targeting gene (T), as well as the *ermF* and *sacB* genes. Erythromycin-resistant (Em-resistant) transformants were subjected to counterselection by cultivation on TS agar plates containing 7% sucrose and 10 µg/mL Erm. Transformants that underwent a double homologous recombination event to remove the vector were selected. (**B**) Chromosomal structures at the *oxyR* loci of the mutant. The Coding Sequences are depicted by arrows. Black arrows indicate target genes. Black triangles indicate PCR primers. (**C**) Agarose gel electrophoresis of the PCR products obtained by the primer pairs indicated on left. Lane 1, OMA14 (wild type); Lane 2, targeting plasmid pND0003; Lane 3, *oxyR::ermF* (V3147, clone A); Lane 4, *oxyR::ermF* (V3147, clone B); and Lane 5, distilled water (negative control). (**D**) Verification of the *oxyR* transcript deletion by RNAseq. Top: Transcript profile in the parental, wild-type strain (V3203), bottom: transcript profile in the OxyR-deficient mutant, V3147. Reads per kilobase million (RPKMs) for *ahpF, ahpC, eno* and *oxyR* are shown in brackets beneath the names of the genes. Read ratio % indicates the percentage of total RNAseq reads.

**Figure 3 microorganisms-09-00551-f003:**
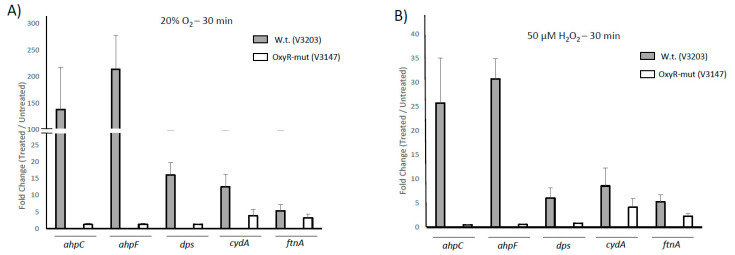
Role of OxyR in regulation of oxidative stress response genes in *P. intermedia*. Regulation of potential ROS stress response genes to H_2_O_2_ and aeration in the wild-type (V3203) and *oxyR* mutant (V3147) OMA strains. Exponentially growing cell cultures were exposed to (**A**) aerobic conditions with vigorous shaking for 30 min or (**B**) two doses of 50 µM H_2_O_2_ (added at time point 0 and at time point 15 min) and the challenged cells were collected at time point of 30 min. The mRNA was purified and quantified via qRT-PCR. Data represent the fold change of treated samples when compared to an untreated control. Error bars represent standard error of the mean of 3 biological replicates.

**Figure 4 microorganisms-09-00551-f004:**
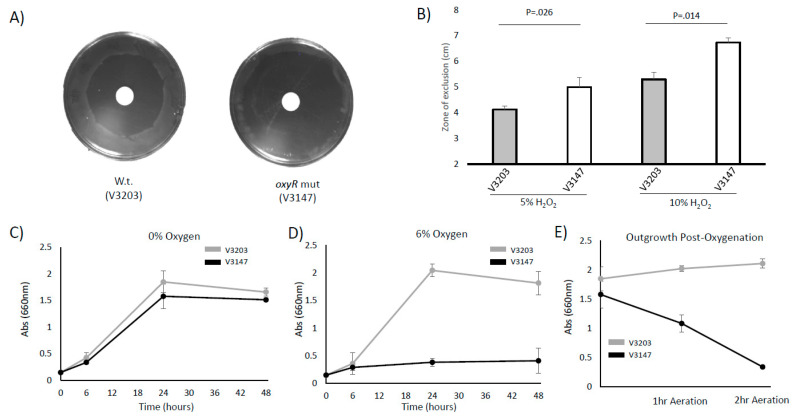
OxyR is required for survival of *P. intermedia* with oxidative stress. (**A**) Wild-type (V3203) and *oxyR* mutant (V3147) OMA strains were plated on anaerobic Tryptic Soy Argar (TSA) plates and exposed to a disc containing 10% hydrogen peroxide. After 48–72 h incubation, the zone of inhibition was measured. (**B**) The zone of inhibition was measured from 3 repliates using a disc containing 5% or 10% hydrogen peroxide. (**C**) Wild-type and *oxyR* mutant OMA strains were grown anaerobically in Brain Hearth Infusion (BHI) broth for 48 h and growth was assessed via OD_660_. (**D**) The growth of the wild-type (V3203) and *oxyR* mutant (V3147) was compared in BHI broth in an atmosphere of 6% oxygen. (**E**) After 1 or 2 h of aeration with vigorous shaking, cells were pelleted and resuspended in anaerobic media normalized to the same starting OD_660_ (0.15). Cells that were not aerated are shown as a comparison (the 0 h mark). Cultures were grown for 24 h and growth recorded as the optical density. All experiments are representative of at least 3 biological replicates. Error bars are the standard error of the sample mean.

**Figure 5 microorganisms-09-00551-f005:**
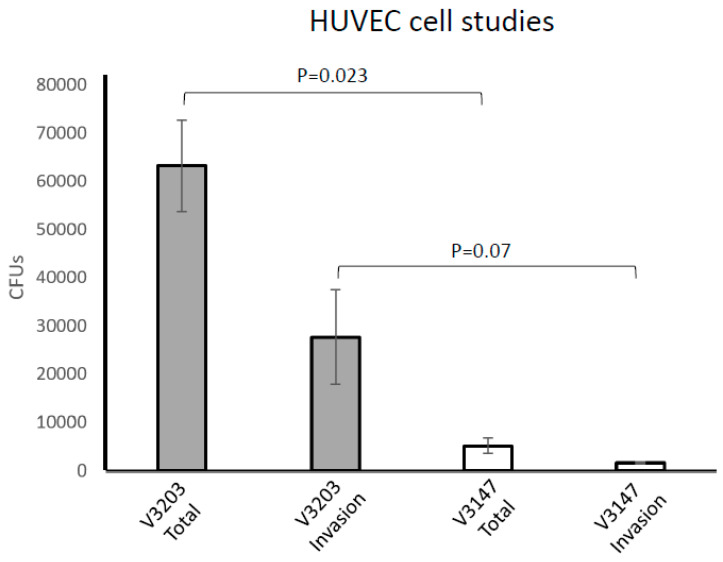
OxyR is required for survival of OMA in host cells. A host cell interaction was performed using the wild-type (V3203) and *oxyR* mutant (V3147) OMA strains. Primary Human Umbilical Vein Endothelial cells (HUVECs) were infected with bacteria at an MOI of 100:1 for 1 h. After infection, host cells were washed and lysed and bacteria were plated on TSA blood agar plates. To quantify invasion, cells were treated with 300 µg/mL metronidazole and 400 µg/mL gentamicin to kill extracellular bacteria before washing and lysing cells. Error bars represent standard error of the mean of 3 biological replicates.

## Data Availability

RNAseq data were deposited in the Gene Expression Omnibus (GEO) with the reference number GSE168003.

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
