# Peer review of "Insertional Inactivation of Prevotella intermedia OxyR Results in Reduced Survival with Oxidative Stress and in the Presence of Host Cells"

_microorganisms, 2021, doi:10.3390/microorganisms9030551_

Round 1

Reviewer 1 Report

This is the first report demonstrating construction of P. intermedia mutant strain. The procedure is properly described. Analysis of the oxyR mutant strain in this species is presented quite good. However, more in-depth description of results obtained and discussion would be helpful.

Main comments:

Page 2, lines 72-75: this information is supported by the reference #16 (analysis in P. intermedia), demonstrating expression of P. gingivalis proteins, HmuY and HmuR. Here, citation on P. gingivalis expression of these proteins should be first included (Lewis et al., Microbiology, 2006, 152, 3367-3382 and Olczak et al., Arch Microbiol, 2008, 189, 197-210; these references show slightly different transcript expression pattern in regard to particular operon regions). Further information on the HmuY homologs expressed by P. intermedia (termed PinO and PinA), produced at higher levels under iron starvation, was reported recently and should be cited (Bielecki et al., Biochem J, 2020, 477, 381-405). This information can be helpful in the transcript expression analysis presented in the manuscript. In Supplementary Table S5, decreased transcript level is shown for PinA protein (indicated as HmuY protein; PIOMA14_RS06100), encoded on the small chromosome. What about the second, more similar HmuY homolog, namely PinO, encoded on the large chromosome ?

Page 3, lines 114-117: it seems that iron and heme starvation used in this study would be not sufficient, taking into consideration that P. intermedia could be able to introduce free iron into the PPIX ring. At least some discussion on choosing such conditions instead of full starvation is required.

Supplementary Tables S5-S8 present detailed analysis of transcript expression, but the data are not discussed and general conclusions are not presented.

Analysis of transcript expression: in Materials and methods, RNAseq analysis: details of cell culturing are required (how long P. intermedia were grown before harvesting ? please correlate this with data shown in Figure 4C). In Quantitative RT-PCR analysis: “Cells were grown to mid-log phase…” – please indicate time (how many hours bacteria were grown, correlate this with the growth curve shown in Figure 4C). Please mention why this time point was used. In RT-PCR analysis, first more than 1 time point of the bacterial growth should be analyzed and shown in the manuscript, demonstrating expression of OxyR at the transcript level in the wild type strain.

Page 4, lines 238-242: anaerobic conditions used to co-culture P. intermedia with model host cells are optimal for bacteria but not for mammalian cells. At least some discussion is required to confirm that host cells can grow properly under these conditions. In addition, how long and what conditions were used to culture P. intermedia before infection ? How bacteria were prepared for infection assay ?

Other comments to the entire manuscript:

- page 1, line 40: should be Lopez et al.

- different font is used throughout the entire manuscript (for example page 3, line111; page 6, lines 285, 286; and in other places)

- different style of “hour” is used (for example 4hr, 1 hour 1 h)

- please distinguish between “similarity” and “identity” – there is a difference in percentage between description in the text and in the legend to Figure 1

- throughout the entire manuscript, very often name of species are not shown in italic.

- Figure 1: should be C. (not C )

- labeling of the figures is dual, such as A) and (a) – this is confusing, especially in the Figure 2 – this is confusing

- description of the mutant: site-directed mutant means that specific site-directed mutagenesis was carried out (such as substitution of nucleotides, insertion or deletion of short regions); deletion mutant strain would be better description for the P. intermedia mutant strain constructed in this study

- Figure 3 legend: should be wild type (is wildtype)

- Style of citation in the Discussion section should be corrected (sometimes there is upper case used sometimes not)

- Please correct the list of references – names of species should be in italic; different style of presentation of references was used – please unify

- In general, the entire manuscript requires thorough proofreading/editing.

Author Response

The response is attached 

Reviewer 2 Report

Naito et al screened one ATCC strain and 30 clinical isolates of P. intermedia for transformation ability and successfully generated a OxyR-deficient OMA14 mutant and revealed that the mutant was very sensitive to the oxygen stress due to downregulation of AhpCF systems. The OxyR mutant also showed weak virulence by severely impairing in its ability to grow with HOKs and HUVECs cells. It is the first study to investigate the P. intermedia genetics and built the foundation for P. intermedia molecular biology study. I have several suggestions for this study as following:

  1. The author didn’t distinguish the conserved catalytic cysteines in Figure 1B.
  2. Figure 3, although cydA and ftnA genes had modest up-regulation, but the expression in mutant was still lower comparing with wild-type, OxyR also effected their expression modestly.
  3. Figure 4C showed that the mutant had impaired growth even under anaerobic condition which means OxyR site might cause strain weaker, it might be one of the reason causing its subdued virulence.
  4. Figure 5, we can see the difference between WT and mutant is huge, it would be better to add the P value or mark with asterisk.
  5. It would be better to find a way to complement the mutation and verify the genetic phenotype of OxyR could be complementary.
  6. Discussion part may need modification in writing format and language.

Author Response

The response is attached 

Reviewer 3 Report

Naito et al Microorganisms 2021 My Review

Prevotella intermidia is a gram negative anaerobic organism that is highly enriched in subgingival plaque in states of periodontitis. As well, this organism is well known to invade oral epithelial cells. To date, like the overwhelming majority of wild bacteria, no strain of Prevotella intermedia has been genetically tractable to insertional mutagenesis. Here, Naito and colleagues report the first genetic deletion in a strain of P. intermedia (a clinical strain they isolated). They deleted the OxyR gene which codes for a protein in P. intermidia that detects oxygen and reactive oxygen species and activates transcription of genes involved in free radical scavenging. The authors demonstrated successful knockout with PCR and gel electrophoresis, then went on to demonstrate that the knockout has reduced fitness in oxygen and H2O2 rich conditions and showed that it has reduced fitness when co-cultured with two primary human cell lines. The manuscript is excellently written. All controls were performed and the data is able to be interpreted. The authors point out that the OxyR gene is a potential target for intervention in progression of periodontitis and all of their data supports that conclusion.

What follows are some minor suggestions to improve the manuscript:

Fig 1b: Are the asterisks that mark conserved cysteines supposed to be red?

L371: What were the conditions of growth when performing comparative RNAseq? Were these grown aerobically? If anaerobic, can the authors speculate on why the OxyR gene changed gene expression levels of other genes?

L524: " In Neisseria the OxyR acts as the repressor of the major anti-oxidative stress response mechanism"

Change "...acts as the repressor" to "...acts in repression"

L528: "... the regulatory mechanism resembles that of OxyR present in other members of the Bacteroidetes group where it mainly functions as the activator of oxidative stress response mechanisms."

Change "...functions as the activator..." to "...functions in activation..."

Author Response

The response is attached 
